# Placenta-Specific Transcripts Containing Androgen Response Elements Are Altered In Silico by Male Growth Outcomes

**DOI:** 10.3390/ijms25031688

**Published:** 2024-01-30

**Authors:** Ashley S. Meakin, Melanie Smith, Janna L. Morrison, Claire T. Roberts, Martha Lappas, Stacey J. Ellery, Olivia Holland, Anthony Perkins, Sharon A. McCracken, Vicki Flenady, Vicki L. Clifton

**Affiliations:** 1Early Origins of Adult Health Research Group, Health and Biomedical Innovation, UniSA: Clinical and Health Sciences, University of South Australia, Adelaide, SA 5000, Australia; janna.morrison@unisa.edu.au; 2Mater Medical Research Institute, The University of Queensland, Brisbane, QLD 4101, Australia; vicki.flenady@mater.uq.edu.au; 3Flinders Health and Medical Research Institute, College of Medicine and Public Health, Flinders University, Adelaide, SA 5042, Australia; melanie.smith@flinders.edu.au (M.S.); claire.roberts@flinders.edu.au (C.T.R.); 4Department of Obstetrics, Gynaecology and Newborn Health, Mercy Hospital for Women, The University of Melbourne, Heidelberg, VIC 3084, Australia; mlappas@unimelb.edu.au; 5Department of Obstetrics and Gynaecology, Monash University, Melbourne, VIC 3168, Australia; stacey.ellery@hudson.org.au; 6The Ritchie Centre, Hudson Institute of Medical Research, Melbourne, VIC 3168, Australia; 7School of Pharmacy and Medical Sciences, Griffith University, Gold Coast, QLD 4222, Australia; o.holland@griffith.edu.au (O.H.); aperkins1@usc.edu.au (A.P.); 8School of Health, University of the Sunshine Coast, Sunshine Coast, QLD 4556, Australia; 9Women and Babies Research, Faculty of Medicine and Health, The University of Sydney, Camperdown, NSW 2050, Australia; sharon.mccracken@sydney.edu.au

**Keywords:** androgen response element, androgen receptor, fetal growth, placenta transcriptome

## Abstract

A birthweight centile (BWC) below the 25th is associated with an elevated risk of adverse perinatal outcomes, particularly among males. This male vulnerability may stem from alterations in placenta-specific androgen signalling, a signalling axis that involves the androgen receptor (AR)-mediated regulation of target genes containing androgen response elements (AREs). In this study, we examined global and ARE-specific transcriptomic signatures in term male placentae (≥37 weeks of gestation) across BWC subcategories (<10th, 10th–30th, >30th) using RNA-seq and gene set enrichment analysis. ARE-containing transcripts in placentae with BWCs below the 10th percentile were upregulated compared to those in the 10th–30th and >30th percentiles, which coincided with the enrichment of gene sets related to hypoxia and the suppression of gene sets associated with mitochondrial function. In the absence of ARE-containing transcripts in silico, <10th and 10th–30th BWC subcategory placentae upregulated gene sets involved in vasculature development, immune function, and cell adhesion when compared to those in the >30th BWC subcategory. Collectively, our in silico findings suggest that changes in the expression of ARE-containing transcripts in male placentae may contribute to impaired placental vasculature and therefore result in reduced fetal growth outcomes.

## 1. Introduction

Disrupted fetal growth can result in a small for gestational age (SGA; <10th birthweight centile (BWC)) neonate who is at increased risk of morbidity and mortality [1], but the contributing molecular mechanisms remain unclear [2,3,4,5]. While SGA is a known risk factor for adverse neonatal outcomes, recent work identified that the probability of adverse outcomes was significantly increased up to the 25th BWC when compared with neonates born between the 25th and 90th BWC, and that male neonates were more likely to have a severe adverse outcome compared to females [6]. These observations favour previous studies describing the male disadvantage hypothesis [7] and emphasise that fetal growth to less than the 25th BWC contributes to adverse outcomes. However, studies characterising molecular pathways associated with reduced fetal growth primarily focus comparisons between the <10th and >10th BWC groups. Based on initial observations in the placenta [7], Clifton and colleagues hypothesised that males prioritise growth over adaptation in utero in the presence of a stressor. However, this male-specific response impairs their ability to adequately respond to shifts in the intrauterine environment such as hypoxic insults and inflammation, which then places them at a greater risk for adverse perinatal outcomes. Despite this, the exact mechanisms that contribute to this male-specific response is unclear. Previous work from our group indicates that male fetal growth outcomes may be mediated through differences in placental androgen signalling regulated by the differential expression of androgen receptor (AR) variants [8].

Multiple AR protein variants have been characterised in human [8] and sheep [9] placenta that challenge the current understanding of the placenta-specific androgen signalling axis. This work showed that placental AR protein variants may influence the regulation of key AR-mediated transcripts, particularly in males, associated with altered growth outcomes. Specifically, we showed that increased nuclear expression of the N-terminally truncated variant, AR-45, was associated with continued male growth and increased angiogenic downstream gene expression [8]. However, in males with growth <10th BWC, this AR-45-specific adaptation was impaired. Indeed, different AR protein variants can initiate transcription through an interaction with the *cis*-regulatory element (androgen response element (ARE)) associated with the AR [10]. These 15 base-pair sequences are composed of two palindromic half sites separated by a three-base nucleotide spacer (i.e., AGAACA_NNN_TGTTCT). ARE half and full sites are important initial factors that drive the transcriptional regulation of genes belonging to the AR-mediated transcriptome. Despite this growing understanding of the AR-mediated transcriptome, its function in the placenta is largely uncharacterised. Thus, in the current study, we characterised global and ARE-specific transcriptomic signatures in male placentae from different BWCs (<10th, 10th–30th, and >30th) and performed gene set enrichment analysis (GSEA) on global and ARE-containing transcripts.

## 2. Results

### 2.1. Maternal and Neonatal Characteristics

No maternal measures were significantly different between the BWC subgroups; however, gestation, BWC, birthweight and placenta weight were significantly different (Table 1).

### 2.2. Placental Transcript Expression Variation in Response to Fetal Growth Outcomes

There were 26 transcripts differentially expressed between >30th and <10th BWC males, and 34 transcripts were differentially expressed between 10th–30th and <10th BWC males (Figure 1A). Three transcripts (*PNCK*, *MXI1*, and *ARNT2*) were commonly differentially expressed in the >30th vs. <10th BWC and 10th–30th vs. <10th BWC comparisons (Figure 1B). *XIST* was commonly differentially expressed in the 10th–30th vs. <10th BWC and >30th vs. 10th–30th BWC comparisons. Twenty-three transcripts were unique to the >30th vs. <10th BWC comparison (1 upregulated, 22 downregulated), and 30 were unique to the 10th–30th vs. <10th BWC comparison (2 upregulated, 28 downregulated; Figure 1B). A complete list of differentially expressed transcripts between the BWC subcategories can be found in Table 2.

### 2.3. Differentially Expressed Transcripts Contain ARE Full and Half Sites

Using a comprehensive and refined ARE dataset previously published [10], differentially expressed transcripts from the current study were integrated to identify whether they contained ARE full (AGAACA_NNN_TGTTCT) and/or half sites, and whether these transcripts were previously reported to be positively or negatively regulated by these ARE sites. Of the 34 transcripts that were differentially expressed in the <10th vs. 10th–30th BWC comparison, 16 contained ARE full sites and 17 contained ARE half sites, two of which have previously been reported to be regulated by both ARE full and half sites (HK2 and SLCO5A1; Table 2). In the <10th vs. >30th BWC comparison, 17 transcripts contained ARE full sites and 21 contained ARE half sites (Table 2); however, none of these have been previously reported to be regulated by either ARE full or half sites.

### 2.4. Gene Set Enrichment Analysis

We performed GSEA on global gene lists, as well as gene lists of transcripts with or without ARE full sites (ARE-positive and ARE-negative, respectively). For global GSEA (see Appendix A), terms upregulated in <10th BWC relative to >30th BWC were primarily involved in hypoxia and vasculature development. Comparatively, terms downregulated in <10th BWC relative to >30th BWC were primarily involved in mitochondrial respiration. Consistent with the <10th BWC relative to >30th BWC comparison, terms involved in hypoxia were upregulated in the <10th BWC relative to the 10th–30th BWC comparison. In >30th BWC placentae, terms involved in protein localisation to phagophore assembly sites and COPII-coated vesicle cargo loading were upregulated when compared with 10th–30th BWC placentae. No terms were downregulated in male >30th BWC vs. 10th–30th BWC.

Given our primary research question focused on understanding placental molecular signatures in males that may be regulated by androgen signalling, we performed subsequent GSEA on ARE-positive and ARE-negative gene lists and compared any findings to male-specific global GSEA terms (Appendix A). Interestingly, in the ARE-negative GSEA, there were 106 unique terms upregulated in the <10th BWC vs. >30th BWC comparison; these terms were primarily involved in cell adhesion and vasculature development (Figure 2A). Similarly, in the ARE-negative GSEA, 460 unique terms were upregulated in the 10th–30th BWC vs. >30th BWC comparison; these terms were primarily involved in immune function and vasculature development (Figure 2C). Twenty-two terms were commonly upregulated in smaller male placentae (<10th and 10th–30th BWC) when compared with >30th BWC placentae, including myogenesis, extracellular matrix structure, and integrin-mediated signalling.

## 3. Discussion

Despite growing evidence that adverse outcomes for the developing male fetus are more likely to occur up to the 25th BWC when compared with those born within the 25th to 90th BWC [6], the underlying molecular mechanisms driving these outcomes remain poorly understood. Herein, we show for the first time that male placenta-specific transcriptomic signatures are altered by distinct fetal growth outcomes. Importantly, GSEA identified that in the absence of transcripts containing ARE full sites, there was increased enrichment in gene sets involved in cell adhesion, vasculature development, and immune function in the <10th and 10th–30th BWC, when compared with the >30th BWC. In line with previous work by our team and others [8,11,12,13,14,15,16], these in silico findings suggest that the increased risk of poor outcomes associated with reduced male fetal growth may be caused, in part, by AR-mediated placental dysfunction that results in perturbations to vasculature development, cell adhesion, and immune function.

Although several of the differentially expressed transcripts in the <10th BWC males contained AREs, only HK2 and SLCO5A1 have previously been reported to be positively or negatively regulated by ARE full sites, respectively. The observed increase in HK2 expression in the <10th when compared with >30th BWC in male placentae is indicative of increased activity of placental androgen signalling. HK2 encodes the hexokinase 2 protein, is involved in the metabolism of glucose [17], and has also been reported to promote the proliferation of certain cancer cells [18,19,20]. The overexpression of AR results in increased HK2 mRNA and protein expression [21]. HK2 expression can also be activated by HIF1α [22,23], which suggests that increased HK2 expression in male placentae <10th BWC is the result of either androgen- or hypoxia-mediated transactivation. Indeed, there is growing evidence that androgen signalling can result in placental hypoxia via a reduction in uterine blood flow [24,25]. Likewise, hypoxia can increase androgen concentrations and enhance AR activity in endocrine tissue, including the prostate [26,27,28,29,30,31]. Thus, the observed increase in HK2 expression in male placentae <10th BWC may be directly regulated by either androgen- or hypoxia-mediated signalling. It is also possible that the enhanced activation of androgen-mediated signalling via a hypoxic in utero environment drives HK2 expression in male placentae <10th BWC. Evidently, further studies are needed to untangle androgen- and hypoxia-dependent signalling pathways in the placenta and define how they contribute to male fetal growth outcomes.

SLCO5A1 expression was also increased in male placentae <10th BWC when compared with male >30th placentae. Whereas HK2 is positively regulated by ARE full sites, SLCO5A1 is reported to be negatively regulated by ARE full sites [10]. SLCO5A1 encodes the solute carrier organic anion transporter family member 5A1 protein. SLCO5A1 not only functions to transport amphipathic molecules across the plasma membrane, but may also function to mediate pathways involved in differentiation and migration [32]. In normal and cancerous prostate tissue, SLCO5A1 expression is significantly reduced in response to androgen deprivation treatment [33], which may be the result of reactivated AR signalling following such treatment. Like HK2, SLCO5A1 expression can be altered by hypoxic insults [34]. Indeed, given that previously published studies report the negative regulation of SLCO5A1 via AR activation, it is likely that alternative pathways such as those regulated by hypoxia are involved in its increased expression in the placentae of small males. Collectively, the observed increase in HK2 and SLCO5A1 expression in male placentae <10th BWC may indicate their direct regulation by either androgen- or hypoxia-mediated signalling. It is important to note that the interpretation of these data is based on comprehensive studies of the AR cistrome in prostate cancer. Given that tissue-specific responses to androgens are reported [35], future studies would benefit from defining the placenta-specific AR cistrome; however, this is beyond the scope of the current study.

Our global GSEA revealed term enrichments consistent with other studies demonstrating that impaired oxygen supply to the fetoplacental unit can result in fetoplacental hypoxia and reduced fetal growth [36]. In <10th BWC males compared with larger males, we identified that upregulated terms were involved in the response to hypoxia, whereas terms that were significantly downregulated were involved in mitochondrial respiration. This finding is supported by previous studies that demonstrate that male-specific alterations to mitochondrial respiration in response to prenatal hypoxia may impair mitochondrial function [37,38]. Our current data suggest that males are less adaptable to intrauterine environments that contribute to reduced fetal growth, as evidenced by disrupted mitochondrial function. This impairment may contribute to an increase in the prevalence of intrauterine morbidity and mortality associated with reduced fetal growth in males [39]. Although androgens are increasingly being recognised as important regulators of mitochondrial function [40], recent work demonstrated that androgen exposure in the trophoblast cell line ACH-3P resulted in mitochondrial dysfunction and enhanced reactive oxygen species (ROS) production [41]. Despite this recent work demonstrating androgen-mediated mitochondrial dysfunction in trophoblast cells, it cannot be concluded that the observed changes to mitochondrial processes in the current study are due to altered androgen signalling in utero. Nonetheless, aberrant ROS production can result in oxidative stress and subsequent placental insufficiency [42], and thus, reduced fetal growth outcomes. Therefore, it is to be expected that an enrichment of terms associated with these processes was reported in <10th BWC males. Indeed, comprehensive studies in the placenta are needed to understand whether androgen-dependent mechanisms contribute to the regulation of mitochondrial function in utero and determine what this means for sex-specific fetal growth outcomes.

Intriguingly, our in silico analysis demonstrated that the absence of ARE-containing transcript results in an enrichment of gene sets involved in vasculature development, cell adhesion, and immune function in small-male placentae (i.e., <10th and 10th–30th BWC) when compared with the placentae males with normal growth (>30th BWC). These findings support growing evidence that functional AR signalling contributes to placental vasculature impairment, and we postulate that this may be due to changes in the expression of AR protein variants. The placentae of pregnancies associated with elevated androgen concentrations during pregnancy have reduced volume and weight and enhanced placental inflammatory pathways and lesions [15,43] when compared with the placentae of healthy controls. In support of these observations, animal studies have demonstrated that elevated androgen concentrations reduce uterine artery blood flow, spiral artery elongation, and placental oxygenation, which, in turn, results in reduced fetal growth [12,24,44]. Our seminal findings of multiple AR protein variants in the human placenta demonstrated that increased androgen concentrations in vitro reduced the nuclear expression of AR-45, whereas the nuclear expression of other measured AR protein variants remained unchanged [8]. AR-45 functions antagonistically to the full-length AR protein variant, is associated with increased angiogenic downstream target gene expression, and may therefore modulate the AR-mediated transcriptomic signature to enhance placental vascular function [45,46]. Thus, impaired placental AR-45 function may enable the dysregulated activity of placenta-specific androgen signalling and exacerbate placental dysfunction and fetal growth perturbations in the presence of chronic intrauterine stressors. Although we did not measure AR-45 protein expression or circulating maternal androgen concentrations in the current cohort due to sample availability, future studies would benefit from examining the associations between androgen ligand concentrations (e.g., testosterone or dihydrotestosterone), AR protein variant expression, and ARE-containing transcript abundance, to identify how any observed interactions vary amongst fetal growth outcomes.

Despite our work highlighting distinct changes to placenta-specific transcriptomic signatures in males from different BWCs, and that changes to gene set enrichment were associated with the presence or absence of ARE-containing transcripts, this study is not without limitations. Indeed, the decision to analyse male placentae only captures the ‘male’ side of an important story of sex differences in placentology. However, the decision to analyse male samples only was hypothesis-driven and based on our previously published works that show that male, but not female, placentae modulate androgen signalling pathways in response to a maternal stressor or reduced fetal growth outcomes [8], or ex vivo inflammatory challenges [11]. Indeed, these studies support our seminal findings that female and male placentae respond differently to the same intrauterine environment [7], and we expect that the molecular adaptations that occur in females across different BWCs would be distinct to those observed in males. Nonetheless, future studies would benefit from interrogating the transcriptomic profiles and molecular regulatory systems that occur in the placentae of small females; however, this is beyond the scope of our current study. Another important consideration is that we used publicly available prostate cancer data [10] to inform us of transcripts that may be regulated by the AR in the placenta, as determined by the presence of an ARE full site. Despite this, while AREs are global motifs, the direction of regulation may differ between tissues. Indeed, similarities between cancer and placenta biology exist [47]; it may therefore be inferred that comparable responses to androgens, as measured by the expression of ARE containing transcripts, between the two also exist. However, given that differences in AR protein variant profiles in placenta and prostate cancer have been observed [8], it is likely that tissue-specific responses to androgens, and thus, changes in the direction of the regulation of AR-mediated transcripts occur. Future studies that apply AR ChIP-seq coupled with RNA-seq studies are therefore needed to define the placenta-specific AR-mediated pathways altered in males of different BWCs; however, this is beyond the scope of the current study. Regardless, our in silico work does provide the rationale to facilitate such future studies.

## 4. Materials and Methods

### 4.1. Sample Cohort

Ethical approval was provided from the Mater Human Research Ethics Committee (53157). Placentae were derived from five biobank sites across Australia based in Melbourne, Brisbane, and Sydney. Each biobank provided term placentae from all BWC categories, and samples were collected and processed following the CoLab protocol as previously described [48]. Samples were matched for maternal BMI and age. For this study, only male placentae with a <10th BWC outcome (*n* = 16) or >10th BWC outcome (*n* = 21) were used. Placentae from >10th BWC outcomes were stratified into two subcategories: 10th–30th BWC (*n* = 12) and >30th BWC (*n* = 9). The decision to use 10th–30th BWC samples was informed by unpublished gene array data from our group.

### 4.2. RNA Extraction and RNA Sequencing

RNA extraction was completed using a QIAGEN RNeasy Mini Kit (QIAGEN, Hilden, Germany) as per the manufacturer’s protocol. Total RNA concentration was recorded using NanoDrop 100 spectrophotometer. The RNA integrity numbers (RIN) for samples used in the initial RNA-Seq analysis were measured using an Agilent RNA 6000 Nano Kit (Agilent, Santa Clara, CA, USA), as per the manufacturer’s protocol. RNA-Seq was performed on an Illumina NovaSeq 6000 (Illumina, San Diego, CA, USA),100 base-pair single-end reads, at the Australian Genomic Research Facility (AGRF), Brisbane.

### 4.3. Data Processing and Differential Gene Expression Analysis

Per-base sequence quality analysis of raw, paired-end sequencing reads was performed using the Rsubread v2.6.4 [49] package. RNA-seq libraries were aligned to the human reference genome (GRCh38.p13) using the Rsubread package. To quantify gene expression, we used featureCounts from the Rsubread package. The resultant count matrix contained expression counts for 43,689 genes for the 76 RNA-seq libraries.

Differential gene expression analysis was performed using R Statistical Software (v4.0.5) using the packages *edgeR* v3.40.0 and *DESeq2* v1.38.0 [50,51]. We retained 23,129 genes with >10 cpm in at least 7 of the 76 RNA-seq libraries. We calculated and applied TMM-normalisation factors and dispersion to account for differences in library size. Differential gene expression was performed using *DESeq2*. Genes were considered differentially expressed if the Benjamini–Hochberg adjusted *p*-value associated with the estimated log2 fold change (log2FC) was below 0.05. Data visualisation was prepared using the *ggplot* v3.4.2 R package [52].

### 4.4. Gene Set Testing

GSEA v4.1.0 [53,54] was used to perform pre-ranked gene set analysis. For each comparison, genes were ranked based on calculating the sign of the log2FC multiplied by the negative log10 FDR. The gene sets used for the analysis were derived from the Molecular Signature Database (Human MSigDB v2023.2Hs) [53,55] and include the Hallmark (h.all.v2022.1.Hs.symbols.gmt), Curated (c2.all.v2022.1.Hs.symbols.gmt), Gene Ontology (c5.all.v2022.1.Hs.symbols.gmt), and Immunologic signatures (c7.all.v2022.1.Hs.symbols.gmt). Briefly, gene sets were filtered using the following parameters: a minimum gene size of 15 and a maximum gene size of 2000 with a weighted enrichment statistic. Gene features with no symbol match were omitted. For each comparison, ranked genes were compared to the gene sets using a GSEA normalised enrichment score (NES) and an adjusted *p*-value. The NES is a normalised enrichment score that measures the degree of enrichment for a gene set in each gene expression dataset. Gene sets were considered significant if they had an adjusted *p*-value of <0.05.

## 5. Conclusions

The current body of work demonstrates distinct differences in global transcriptomic signatures that may contribute to altered placental function and fetal growth outcomes in males. Indeed, the data presented herein may suggest activated AR signalling in the placentae of males with reduced birthweight (<10th or 10th–30th BWC). We show that the absence of ARE-containing transcripts results in the enrichment of terms involved in vasculature development, cell adhesion, and immune function in silico. These findings are supported by previous in vitro and in vivo studies that show that the activation of AR signalling pathways in the placenta can impair its function and result in reduced fetal growth outcomes. Therefore, comprehensive characterisation studies of the AR-mediated signalling axis are needed to validate the current in silico work, and to identify novel targets to modulate this pathway that result in improved placental function and fetal growth outcomes, thereby reducing adverse perinatal outcomes, particularly for males.

## Figures and Tables

**Figure 1 ijms-25-01688-f001:**
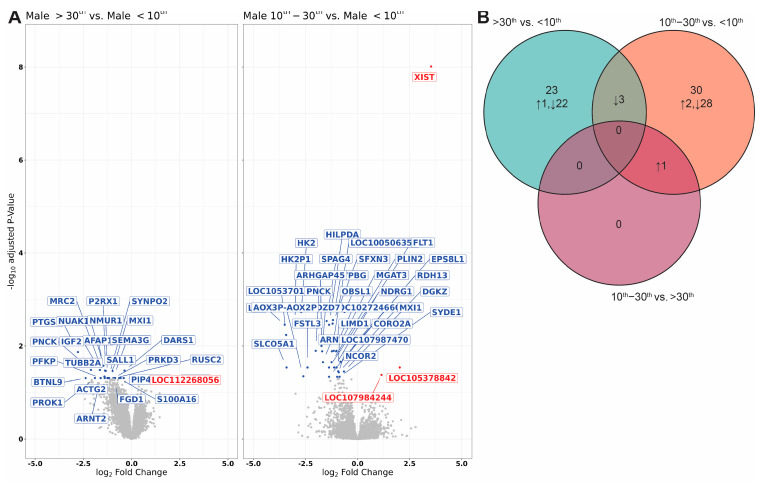
Differentially expressed transcripts from males of different BWC outcomes. (**A**) Volcano plot visualisation of differentially expressed transcripts from the >30th BWC vs. <10th BWC and the 10th–30th vs. <10th BWC comparisons. Red data points represent transcripts that were significantly upregulated and blue data points represent transcripts that were significantly downregulated in >30th or 10th–30th BWC males compared with <10th BWC males; grey data points represent non-significant transcripts. (**B**) Venn diagram representing differentially expressed placental transcripts (↑ = upregulated; ↓ = downregulated) between males with >30th, 10th–30th, and <10th BWC outcomes. An adjusted *p*-value of 0.05 was used to identify transcripts that were differentially expressed between the BWC subcategories of male placentae.

**Figure 2 ijms-25-01688-f002:**
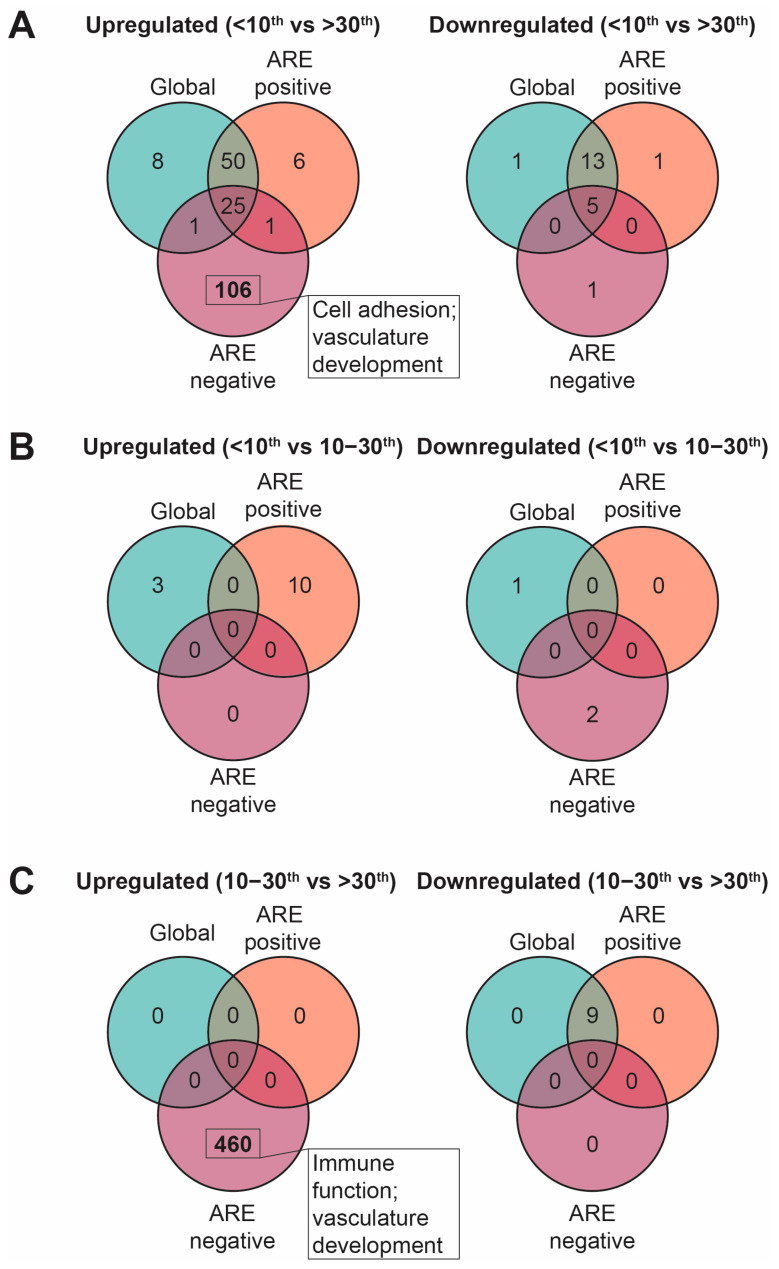
Venn diagram visualisation of upregulated and downregulated GSEA terms in (**A**) <10th vs. >30th, (**B**) <10th vs. 10th–30th, and (**C**) 10th–30th vs. >30th. ARE positive = transcripts containing only AREs; ARE negative = transcripts without AREs.

**Table 1 ijms-25-01688-t001:** Maternal and neonatal characteristics.

	BWC			
	<10th	10th–30th	>30th	
	*n* = 16	*n* = 12	*n* = 9	*p*
Maternal age (yrs)	32.0 (29.5–34.5)	33.5 (30.0–35.0)	32.0 (29.0–33.0)	0.638
BMI	20.9 (19.3–21.6)	26.5 (21.9–33.2)	21.6 (21.1–31.5)	0.097
Parity	1.0 (0.0–1.0)	2.0 (1.0–2.0)	1.0 (1.0–2.0)	0.106
Gestation (days)	262.5 (260.5–267.0) ^a^	277.5 (268.0–283.5) ^b^	274.0 (267.0–278.0) ^b^	**0.004**
BWC (%)	2.5 (2.5–5.2) ^a^	20.0 (17.6–25.1) ^b^	91.1 (79.2–95.6) ^b^	**0.000**
Birthweight (g)	2468.0 (2232.5–2542.5) ^a^	3225.0 (3035.0–3375.0) ^b^	3900.0 (3760.0–4196.0) ^b^	**0.000**
Placenta weight (g)	520.0 (380.0–576.4) ^a^	550.0 (510.0–592.0) ^ab^	729.5 (600.0–890.0) ^b^	**0.000**
% Asthma	12.5	16.7	11.1	-
% GDM	6.3	0	0	-
% Pre-eclampsia	6.3	0	0	-

Bolded values represent a significant effect from the test variable; different superscripts represent a significant pairwise comparison post hoc test between them. BMI = body mass index; BWC = birthweight centile; GDM = gestational diabetes mellitus.

**Table 2 ijms-25-01688-t002:** Differentially expressed transcripts between male BWC subcategories contain ARE full and half sites.

BWC	Transcript	log2 Fold Change (Relative to <10th)	Adjusted *p*-Value	ARE Full Site	ARE Half Site
10th–30th	*LEP*	−3.50	3.49 × 10^−3^	✓	✓
	*LOC105370135*	−3.43	5.79 × 10^−3^	×	×
	*SLCO5A1* ^1^	−3.41	2.91 × 10^−2^	✓	✓
	*HK2P1*	−3.01	3.49 × 10^−3^	×	×
	*HK2* ^1^	−2.70	1.84 × 10^−3^	✓	✓
	*AOX3P-AOX2P*	−2.60	4.50 × 10^−2^	×	×
	*FSTL3*	−2.40	2.91 × 10^−2^	✓	×
	*PNCK*	−2.01	1.27 × 10^−2^	×	×
	*ARHGAP45*	−1.74	9.99 × 10^−3^	×	×
	*PDZD7*	−1.71	1.30 × 10^−2^	×	×
	*ARNT2*	−1.67	2.23 × 10^−2^	×	✓
	*SPAG4*	−1.49	2.90 × 10^−3^	✓	✓
	*FLT1*	−1.37	2.91 × 10^−2^	✓	✓
	*CORO2A*	−1.37	4.61 × 10^−2^	✓	✓
	*HILPDA*	−1.37	3.49 × 10^−3^	✓	✓
	*PLIN2*	−1.24	2.23 × 10^−2^	×	×
	*LOC100506358*	−1.23	1.30 × 10^−2^	×	×
	*SFXN3*	−1.20	3.30 × 10^−3^	×	×
	*TPBG*	−1.18	2.76 × 10^−3^	✓	✓
	*LOC102724660*	−1.14	1.27 × 10^−2^	×	×
	*NDRG1*	−1.12	2.91 × 10^−2^	✓	✓
	*MGAT3*	−1.03	1.30 × 10^−2^	✓	✓
	*LIMD1*	−1.01	1.27 × 10^−2^	✓	✓
	*RH13*	−1.01	2.91 × 10^−2^	×	×
	*NCOR2*	−0.98	4.61 × 10^−2^	✓	✓
	*LOC107987470*	−0.93	3.51 × 10^−2^	×	×
	*MXI1*	−0.90	3.69 × 10^−2^	✓	✓
	*SYDE1*	−0.87	4.61 × 10^−2^	×	×
	*EPS8L1*	−0.82	2.19 × 10^−2^	✓	✓
	*DGKZ*	−0.66	3.55 × 10^−2^	×	✓
	*OBSL1*	−0.63	1.84 × 10^−3^	×	✓
	*LOC107984244*	1.15	4.19 × 10^−2^	×	×
	*LOC105378842*	2.02	2.91 × 10^−2^	×	×
	*XIST*	3.53	9.76 × 10^−9^	✓	×
>30th	*PTGS2*	−2.78	1.35 × 10^−2^	✓	✓
	*BTNL9*	−2.38	4.92 × 10^−2^	×	✓
	*PNCK*	−2.10	3.29 × 10^−2^	×	×
	*PROK1*	−1.90	4.92 × 10^−2^	✓	✓
	*IGF2*	−1.65	3.29 × 10^−2^	✓	✓
	*ACTG2*	−1.61	4.92 × 10^−2^	✓	✓
	*ARNT2*	−1.61	4.92 × 10^−2^	×	✓
	*PFKP*	−1.60	4.92 × 10^−2^	✓	✓
	*AFAP1*	−1.46	2.67 × 10^−2^	✓	✓
	*MRC2*	−1.41	4.55 × 10^−2^	✓	✓
	*TUBB2A*	−1.40	4.92 × 10^−2^	×	✓
	*NUAK1*	−1.38	3.39 × 10^−2^	✓	✓
	*P2RX1*	−1.35	3.45 × 10^−2^	×	×
	*NMUR1*	−1.26	4.92 × 10^−2^	×	✓
	*SYNPO2*	−1.21	4.92 × 10^−2^	✓	✓
	*SALL1*	−1.17	4.92 × 10^−2^	✓	✓
	*MXI1*	−0.99	3.46 × 10^−2^	✓	✓
	*PRKD3*	−0.94	4.92 × 10^−2^	✓	×
	*SEMA3G*	−0.94	4.92 × 10^−2^	×	✓
	*RUSC2*	−0.88	4.92 × 10^−2^	✓	✓
	*FGD1*	−0.85	4.92 × 10^−2^	✓	✓
	*S100A16*	−0.65	4.92 × 10^−2^	✓	✓
	*PIP4K2A*	−0.56	4.92 × 10^−2^	✓	✓
	*MYH9*	−0.41	4.92 × 10^−2^	✓	✓
	*DARS1*	−0.37	3.39 × 10^−2^	×	×
	*LOC112268056*	0.79	4.92 × 10^−2^	×	×

^1^ = transcripts regulated by ARE half and/or full sites.

## Data Availability

The data generated and analysed during this study are available from the corresponding author on reasonable request.

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
