# Peer review of "Placenta-Specific Transcripts Containing Androgen Response Elements Are Altered In Silico by Male Growth Outcomes"

_ijms, 2024, doi:10.3390/ijms25031688_

Round 1

Reviewer 1 Report

Comments and Suggestions for Authors

Your study is about placental specific signatures in male growth outcomes. 

The LBW newborn may be preterm or small for gestational age (SGA); however, most studies have not distinguished between these two subgroups when evaluating outcome.

In obstetrics the 10% centile is usually used for defining risk hoever I do understand your 25% threshold.

Discussions and conclusions are well documented.

However there are some issues to be addressed:

- in title and abstract section you mentioned that AM signatures are altered by male growth outcomes. You need to be more specific and include the androgen response.

- the topic in relevant in the field but conclusion and discussion section should contain a take home message regarding how your study could be user in clinical practice - eg - prevention of LBW

- we do know that male infants are more difficult to recover after PTB and an underlying mechanism is welcomed but a correlation between these two needs to be clearly mentioned.

- if you did not included maternal testosterone in your data than a mention related to this should be included.

Finally, your direction of study is well documented but limitations and further studies in the field should be clearly mentioned.

Author Response

Your study is about placental specific signatures in male growth outcomes. The LBW newborn may be preterm or small for gestational age (SGA); however, most studies have not distinguished between these two subgroups when evaluating outcome.

Thank you for raising this important consideration. Initial principal component analyses were performed to determine whether such factors (i.e., gestational age, birthweight centiles (BWCs)) impacted placental transcriptomic signatures. Using this approach, we found that BWCs but not gestational age impacted signatures and therefore did not perform subsequent bioinformatic analyses stratified by gestational age. Another important consideration for the current study is that all placental samples were collected from term pregnancies only. This was intentional to reduce any confounding factors associated with preterm delivery that may impact placental-specific transcriptomic signatures.

In obstetrics the 10% centile is usually used for defining risk however I do understand your 25% threshold.

We agree that in obstetrics the 10th percentile is generally used for defining risk; however, “The decision to use 10th-30th BWC samples was informed by unpublished gene array data from our group” (Lines 82-83) that identified enrichment of gene expression in the 10-30th BWC subcategory when compared with <10th and >30th (data unpublished).

In addition, the 25th BWC threshold was based on work published by Yu et al. (PMID 29257760) that showed “…the probability of adverse outcomes was significantly increased up to the 25th BWC when compared with neonates born between the 25th and 90th BWC, and that male neonates were more likely to have a severe adverse outcome compared to females.” (Lines 38-41).

Discussions and conclusions are well documented. However there are some issues to be addressed:

- in title and abstract section you mentioned that AM signatures are altered by male growth outcomes. You need to be more specific and include the androgen response.

Thank you for this feedback. We have updated the abstract to include some detail on the androgen response elements.

- the topic in relevant in the field but conclusion and discussion section should contain a take home message regarding how your study could be user in clinical practice - eg - prevention of LBW

While this is beyond the scope of our current body of work, the data presented herein provides the rationale to explore the complex nature of AR-mediated signalling pathways in the placenta, to understand how it is regulated, and identify whether dysfunction of this pathway can be restored to improve placental function and fetal growth outcomes. We are of the opinion that advancing the fundamental knowledge of androgen signalling in the human placenta is first needed before any clinical translation can be met.  

We thank the reviewer for this suggestion and have amended our final concluding sentence to the following:

Lines 310-314: “Therefore, comprehensive characterisation studies of the AR-mediated signalling axis are needed to identify novel targets to modulate this pathway that result in improved placental function and fetal growth outcomes, thereby reducing adverse perinatal outcomes particularly for males.”

- we do know that male infants are more difficult to recover after PTB and an underlying mechanism is welcomed but a correlation between these two needs to be clearly mentioned.

We agree that male neonates are more likely to experience adverse events after preterm birth; however, this study investigated androgen mediated transcriptomic signatures in term placentae only. Indeed, future work beyond the scope of the current study would benefit from applying a similar workflow in male preterm placentae of different BWCs to determine whether the changes observed in the current study manifest earlier in gestation.

- if you did not included maternal testosterone in your data than a mention related to this should be included.

We thank both reviewers for bringing this important point to our attention.

We were limited in sample availability to measure maternal testosterone concentrations at term delivery; however, we recognise that this is an important consideration that should be included in future study designs. We have, however, amended our discussion to recognise this limitation:

“Although we did not measure AR-45 protein expression or circulating maternal androgen concentrations in the current cohort due to sample availability, future studies would benefit from examining the associations between androgen ligand concentrations (e.g., testosterone or dihydrotestosterone), AR protein variant expression, and ARE containing transcript abundance, to identify how any observed interactions vary amongst fetal growth outcomes.” (Lines 276-282).

Finally, your direction of study is well documented but limitations and further studies in the field should be clearly mentioned.

We agree and had initially included limitations and future direction within the manuscript, that has since been built on:

“Although we did not measure AR-45 protein expression or circulating maternal androgen concentrations in the current cohort due to sample availability, future studies would benefit from examining the associations between androgen ligand concentrations (e.g., testosterone or dihydrotestosterone), AR protein variant expression and ARE containing transcript abundance, to identify how any observed interactions vary amongst fetal growth outcomes.

Despite our in silico work highlighting an important role for androgens in modulating male placental function, this study is not without limitations. Indeed, the decision to ana-lyse male placentae only captures the ‘male’ side of an important sex differences in placentology story. However, the decision to analyse male samples only was hypothesis driven and based on our previously published works that show male, but not female, placentae modulate androgen signalling pathways in response to a maternal stressor or reduced fetal growth outcomes[8], or ex vivo inflammatory challenges[50]. Indeed, these studies support our seminal findings that female and male placentae respond differently to the same intrauterine environment[7], and we expect that the molecular adaptations that occur in females across different BWCs would be distinct to those observed in males. Nonetheless, future studies would benefit from interrogating the transcriptomic profiles and molecular regulatory systems that occur in placentae of small females; however, this is beyond the scope of our current study. Another important consideration is that we used publicly available prostate cancer data[10] to inform us of transcripts that may be regulated by the AR in the placenta, as determined by the presence of an ARE full site. Indeed, AREs are global motifs; however, the direction of regulation may differ between tissues. Future studies would benefit from comprehensive AR ChIP-seq coupled with RNA-seq studies to define the placental-specific AR-mediated pathways altered in males of different BWCs; however, this is beyond the scope of the current study. Regardless, our in silico work does provide the rationale to facilitate such future studies.” (Lines 276-302).

Reviewer 2 Report

Comments and Suggestions for Authors

Meakin and colleagues report on a study designed to characterize AR-mediated transcriptome signatures in term male placenta collected at different birthweight centiles (BWC); <10th, 10-30th, and >30th, using RNAseq and gene set enrichment. Overall the manuscript is well-written and the collected RNAseq data is intriguing and will be of interest to the scientific community. However, there is a major concern regarding the interpretation of the data and conclusions.

The greatest concern relates to the title and wording regarding AR-mediated gene transcription. For example;

-) were maternal testosterone samples collected and correlated with these placentas and BWC

-) what androgen response element sequence was used and included in the data analysis? The material and Method section is missing information how the gene sets were obtained

-) The DNA response element for AR and glucocorticoid receptor is very similar/almost identical. Considering the placenta expressed both AR and GR how can the authors be confident that these are not regulated by glucocorticoids?

-) In order to designate this RNAseq derived data set as AR regulated, an in vitro experiment could be performed using trophoblast cells treated with testosterone with and without the presence of an AR antagonist (flutamide), and selected genes could be examined by qPCR as a measure of validation. And/or compare expression of identified genes in this data set in female (XX) placentas.

Currently, it is unclear based on the material/methods and information provided how these are AR-mediated transcripts, and not just transcripts that correspond to BWC.

Author Response

Meakin and colleagues report on a study designed to characterize AR-mediated transcriptome signatures in term male placenta collected at different birthweight centiles (BWC); <10th, 10-30th, and >30th, using RNAseq and gene set enrichment. Overall the manuscript is well-written and the collected RNAseq data is intriguing and will be of interest to the scientific community. However, there is a major concern regarding the interpretation of the data and conclusions.

The greatest concern relates to the title and wording regarding AR-mediated gene transcription. For example;

-) were maternal testosterone samples collected and correlated with these placentas and BWC

We thank both reviewers for bringing this important point to our attention.

We were limited in sample availability to measure maternal testosterone concentrations at term delivery; however, we recognise that this is an important consideration that should be included in future study designs. We have, however, amended our discussion to recognise this limitation:

“Although we did not measure AR-45 protein expression or circulating maternal androgen concentrations in the current cohort due to sample availability, future studies would benefit from examining the associations between androgen ligand concentrations (e.g., testosterone or dihydrotestosterone), AR protein variant expression, and ARE containing transcript abundance, to identify how any observed interactions vary amongst fetal growth outcomes.” (Lines 276-282).

Nonetheless, we are adapting previously published ChIP-seq datasets obtained from androgen-dependent prostate cancer studies to determine how AR-mediated transcripts in the placenta are altered in response to different male growth outcomes. This in silico approach has provided rationale to assess the relationship between maternal testosterone concentrations, placental-specific AR-mediated transcripts, and placental AR protein expression.

-) what androgen response element sequence was used and included in the data analysis?

The reference list of genes containing AREs was extracted from previously published work that refined the nucleotide signature of the ARE based on ChIP-Seq experiments; the majority of these AREs were annotated as intergenic (50.9%) and intronic (43.5%) regions, which agrees with other ChIP-Seq studies using AR antibodies (PMID: 25183411).

The material and Method section is missing information how the gene sets were obtained

Regarding your query about the gene sets used in our Gene Set Enrichment Analysis (GSEA), we realize the oversight in not explicitly detailing their source.

We have updated the materials and methods section to include more information on how gene sets were obtained. Lines 106-109: “The gene sets used for the analysis were derived from the gene set database including the Hallmark (h.all.v2022.1.Hs.symbols.gmt), Curated (c2.all.v2022.1.Hs.symbols.gmt), Gene Ontology (c5.all.v2022.1.Hs.symbols.gmt), and Immunologic signatures (c7.all.v2022.1.Hs.symbols.gmt).”

-) The DNA response element for AR and glucocorticoid receptor is very similar/almost identical. Considering the placenta expressed both AR and GR how can the authors be confident that these are not regulated by glucocorticoids?

We thank the reviewer for this important consideration and note in the current study’s design that we cannot definitively state that the observed changes in vascular development, immune function, and hypoxia response are exclusively due to the AR. Rather, we state that this is potentially due to the ARE, based on the publicly available ChIP-seq data that comprehensively characterised ARE-containing transcripts, and recognise that other factors, including those regulated by hypoxia, are likely at play.

-) In order to designate this RNAseq derived data set as AR regulated, an in vitro experiment could be performed using trophoblast cells treated with testosterone with and without the presence of an AR antagonist (flutamide), and selected genes could be examined by qPCR as a measure of validation.

We agree with this comment. As previously mentioned, this in silico study provides the rationale to further explore the molecular mechanisms regulated by AR in the placenta. The suggested in vitro work to untangle said molecular mechanisms would be highly impactful in the field of both placental physiology and molecular endocrinology but is beyond the scope of the current study.

And/or compare expression of identified genes in this data set in female (XX) placentas.

The decision to analyse male placentae only was hypothesis driven, as our previous work showed a male-specific adaptation in AR-signalling pathways in response to a maternal stressor or changes in fetal growth (PMID: 31103062).

Currently, it is unclear based on the material/methods and information provided how these are AR-mediated transcripts, and not just transcripts that correspond to BWC

As previously mentioned, we applied published datasets from AR ChIP-seq studies in prostate cancer to identify transcripts in the placenta that contain ARE full-sites. Indeed, based on your previous comments, future in vitro or ex vivo studies that treat trophoblasts or placental tissue with androgens +/- an AR antagonist such as flutamide/enzalutamide or AR-targeted siRNA would provide a comprehensive list of placenta-specific AR mediated transcripts; however, this is well beyond the scope and feasibility of the current study. Importantly, we do note that using a reference list derived from prostate cancer is a limitation (lines 260-263):

‘It is important to note that the interpretation of this data is based on comprehensive studies of the AR-mediated cistrome in prostate cancer. Given tissue-specific responses to androgens are reported [35], future studies would benefit from defining the placental-specific AR-mediated cistrome; however, this is beyond the scope of the current study.’

Round 2

Reviewer 2 Report

Comments and Suggestions for Authors

I appreciate the authors responses to my previous comments. A number of issues remain.

-) please indicate in this study the specific ARE sequences used/included in this study to "identify AR regulated genes".

-) Considering the reference list of genes adapted for this study come from ChIP-Seq data sets from androgen dependent prostate cancer studies, more emphasis needs to be put on specific statements that (1) these are not placenta-specific as the title suggests, and (2) that it is assumed that the genes identified in this study are also regulated by androgens. Without any validation that any of these genes in fact are directly regulated by AR in the placenta, one can only infer and suggest that these genes are regulated by AR in the placenta. Please change the title accordingly and make changes throughout the manuscript indicating that genes highlighted in the study are associated with male BWC. This is the only conclusion that can be drawn based on the current experimental design, and without any validation. In the discussion one can infer that these genes potential are regulated by AR based on comparison to data sets used from previous studies in prostate cancer. In its current form the manuscript is misleading as it suggests placental-specific AR-regulated genes are identified.

-) please provide a link/reference that allows access to these gene data sets ("h.all.v2022.1.Hs.symbols.gmt"???)

Author Response

I appreciate the authors responses to my previous comments. A number of issues remain.

We thank the reviewer for their additional input and have updated the manuscript throughout.

-) please indicate in this study the specific ARE sequences used/included in this study to "identify AR regulated genes".

This has been amended.

Lines 158-163: “Using a comprehensive and refined ARE dataset previously published by Wilson, Qi [10], differentially expressed transcripts from the current study were integrated to identify whether they contained ARE full (AGAACANNNTGTTCT) and/or half sites, and whether these transcripts were previously reported to be positively or negatively regulated by these ARE sites.”

-) Considering the reference list of genes adapted for this study come from ChIP-Seq data sets from androgen dependent prostate cancer studies, more emphasis needs to be put on specific statements that (1) these are not placenta-specific as the title suggests, and (2) that it is assumed that the genes identified in this study are also regulated by androgens. Without any validation that any of these genes in fact are directly regulated by AR in the placenta, one can only infer and suggest that these genes are regulated by AR in the placenta.

We thank the reviewer for their feedback on these two specific points. We agree that the in silico analyses conducted are limited. We also agree that our approach does not allow us to definitely conclude that AR signalling pathways in the placenta are causing the observed changes to male fetal growth outcomes. We have updated our manuscript throughout to reflect these limitations.

Please change the title accordingly and make changes throughout the manuscript indicating that genes highlighted in the study are associated with male BWC.

We thank the reviewer for this suggestion and have amended the title to “Placental-specific transcripts containing androgen response elements are altered in silico by male growth outcomes”.  

This is the only conclusion that can be drawn based on the current experimental design, and without any validation. In the discussion one can infer that these genes potential are regulated by AR based on comparison to data sets used from previous studies in prostate cancer. In its current form the manuscript is misleading as it suggests placental-specific AR-regulated genes are identified.

We agree and thank the reviewer for raising this point for consideration. We have reworked our discussion to reflect the limitations of the study design, highlighting that the findings conducted by the in silico GSEA provide the rationale to conduct comprehensive characterisation studies of AR signalling in human placenta of different BWCs. The below lists changes to sections of the discussion, which we believe is a more appropriate interpretation of the data.

Lines 211-215:

“In line with previous work by our team and others[8, 19-24], these in silico findings suggest that the increased risk of poor outcomes associated with reduced male fetal growth may be, in part, via AR-mediated placental dysfunction that results in perturbations to vasculature development, cell adhesion, and immune function.”

Lines 230-234:

“It is also possible that enhanced activation of androgen-mediated signalling, via a hypoxic in utero environment, drives HK2 expression in male placentae <10th BWC. Evidently, further studies are needed to untangle androgen- and hypoxia-dependent signalling pathways in the placenta and define how they contribute to male fetal growth outcomes.”

Lines 268-280:

“Despite this recent work demonstrating androgen-mediated mitochondrial dysfunction in trophoblast cells, it cannot be concluded that the observed changes to mitochondrial processes are due to enhanced androgen signalling in utero. Nonetheless, aberrant ROS production can result in oxidative stress, subsequent placental insufficiency [44], and thus reduced fetal growth outcomes; therefore, it is to be expected that an enrichment of terms associated with these processes were reported in males <10th BWC. Indeed, further comprehensive studies in the placenta are needed to understand whether androgen-dependent mechanisms contribute to the regulation of mitochondrial function in utero and determine what this means for sex-specific fetal growth outcomes.”

Lines 308-311:

“Despite our work highlighting distinct changes to placental-specific transcriptomic signatures in males from different BWCs, and that changes to gene set enrichment was associated with the presence or absence of ARE-containing transcripts, this study is not without limitations.”

Lines 325-337:

“Despite this, while AREs are global motifs the direction of regulation may differ between tissues. Indeed, similarities between cancer and placenta biology exist[55]: it may therefore be inferred that comparable responses to androgens, as measured by the expression of ARE containing transcripts, between the two also exist. However, given that differences in AR protein variant profiles in placenta and prostate cancer have been observed[8],  it is likely that tissue-specific responses to androgens and thus changes to the direction of regulation of AR-mediated transcripts occur. Future studies that apply AR ChIP-seq coupled with RNA-seq studies are therefore needed to define the placental-specific AR-mediated pathways altered in males of different BWCs; however, this is beyond the scope of the current study.”

-) please provide a link/reference that allows access to these gene data sets ("h.all.v2022.1.Hs.symbols.gmt"???)

The reference for “h.all.v2022.1.Hs.symbols.gmt” is included in text (see reference 18, DOI: 10.1016/j.cels.2015.12.004). RNA-seq data is available through The University of Queensland Research Data Management storage site. Data can be provided with permission from the corresponding author. Please email [email protected] for access.

Round 3

Reviewer 2 Report

Comments and Suggestions for Authors

My comments have been been addressed. Thank you.